# German GPs’ Self-Perceived Role in the COVID-19 Pandemic: Leadership, Participation in Regional Services and Preferences for Future Pandemic Preparedness

**DOI:** 10.3390/ijerph20126088

**Published:** 2023-06-09

**Authors:** Simon Kugai, Dorothea Wild, Yelda Krumpholtz, Manuela Schmidt, Katrin Balzer, Astrid Mayerböck, Birgitta Weltermann

**Affiliations:** 1Institute of General Practice and Family Medicine, University Hospital Bonn, University of Bonn, Venusberg-Campus 1, 53127 Bonn, Germany; 2Nursing Research Unit, Institute for Social Medicine and Epidemiology, University of Lübeck, Ratzeburger Allee 160, 23562 Luebeck, Germany; 3uzbonn, Survey Center Bonn—Center for Empirical Social Research and Evaluation, Oxfordstraße 15, 53111 Bonn, Germany

**Keywords:** general practitioners, pandemic preparedness, COVID-19, regional healthcare, healthcare services, leadership, nationwide survey

## Abstract

General practitioners (GPs) played a vital role during the COVID-19 pandemic. Little is known about GPs’ view of their role, leadership, participation in regional services and preferences for future pandemic preparedness. This representative study of German GPs comprised a web-based survey and computer-assisted telephone interviewing (CATI). It addressed GPs’ satisfaction with their role, self-perceived leadership (validated C-LEAD scale), participation in newly established health services, and preferences for future pandemic preparedness (net promotor score; NPS; range −100 to +100%). Statistical analyses were conducted using Spearman’s correlation and Kruskal–Wallis tests. In total, 630 GPs completed the questionnaire and 102 GPs the CATI. In addition to their practice duties, most GPs (72.5%) participated in at least one regional health service, mainly vaccination centres/teams (52.7%). Self-perceived leadership was high with a C-LEAD score of 47.4 (max. 63; SD ± 8.5). Overall, 58.8% were not satisfied with their role which correlated with the feeling of being left alone (r = −0.349, *p* < 0.001). 77.5 % of respondents believed that political leaders underestimated GPs’ potential contribution to pandemic control. Regarding regional pandemic services, GPs preferred COVID-19 focus practices (NPS +43.7) over diagnostic centres (NPS −31). Many GPs, though highly engaged regionally, were dissatisfied with their role but had clear preferences for future regional services. Future pandemic planning should integrate GPs’ perspectives.

## 1. Introduction

General practitioners (GPs) played a vital role in managing and sustaining healthcare during the COVID-19 pandemic [1,2,3]. More than 90% of all German patients with confirmed or suspected COVID-19 infections were treated by GPs [4]. Studies from other countries confirm the importance of primary care [5,6] and underline its capability to act even with limited resources [1]. However, GPs experienced multiple barriers to fulfilling their role [3], among them poor coordination with other institutions [7,8], lack of resources and guidelines [9,10], as well as lack of inclusion in regional pandemic task forces/networks [11]. This lack of involvement is of particular concern since cooperation between hospitals and primary care may lead to more effective care by containing viral spread and managing patients [12]. A recent study in six different countries found that involving primary care in pandemic planning improved the overall healthcare management during the COVID-19 pandemic [13]. Although GPs’ participation in patient care and additional services were well reported by public media, scientific studies of GPs’ leadership are rare.

In many countries, GPs had to respond to numerous new and adapted services in order to fight the pandemic while sustaining primary care. In Germany, regional differences were observed regarding pandemic-related health services. Examples of new services included stand-alone diagnostic centres and dedicated COVID-19 diagnostic practices [14,15,16], ‘Corona-Taxis’ for driving physicians to quarantined COVID-19 patients [17,18], specialised practices (COVID-19-focus practices) and outpatient treatment units (COVID-19 outpatient clinics/Corona contact points) [19,20,21,22] as well as dedicated COVID-19 vaccination centres [19,20]. Previous studies of GPs’ perception of their role in the pandemic and pandemic processes focused on factors for distress and wellbeing [21], availability of personal protective equipment [10], improved patient flow and practice management strategies [7] as well as testing and job performance [22]. However, little is known about the interplay of GPs’ view of their own role, their role satisfaction, participation in regional health services, and their preferences for future pandemic management.

Given GPs’ role as leaders at the heart of healthcare [3], analysing their contributions to pandemic management is crucial to be better prepared for the next pandemic. Using a web-based survey and telephone interviews, we studied GPs’ views regarding their role, self-perceived leadership, participation in newly established health services and preferences for future pandemic preparedness.

## 2. Materials and Methods

The research was conducted as part of the project egePan Unimed from the nationwide Network of University Medicine (NUM) which is funded by the Federal Ministry for Education and Research. The project described here consisted of a mixed-method study with a web-based survey and computer-assisted telephone interviewing (CATI). It was performed in spring 2021 one year into the pandemic when COVID vaccination campaigns were established nationwide. The survey follows the structure of the German primary healthcare system with a majority of GPs owning single general practices (*n* = 26,784 [2021]) compared to group practices (*n* = 8699 [2021]) and mainly GP-owned ambulatory healthcare centres (*n* = 4179 [2021]) [23].

### 2.1. Sampling

A multilevel clustered randomised sample of GPs was drawn from all active GPs in Germany with a valid e-mail address. The list of all working GPs in Germany was obtained from ArztData AG, a specialised provider for physician addresses. The sample was created in two steps. First, data were stratified in quartiles by federal, state, and regional density. For the 16 German states, 64 county layers were created, and 40% of counties were randomly drawn from each layer. Second, each cluster was stratified into four layers by practice type and nature of employment (‘GP in own practice’, ‘employed in a practice’, ‘director of an ambulatory healthcare centre’, ‘employed in an ambulatory healthcare centre’). For each layer, 30% of GPs were randomly selected and invited to participate in the web-based survey. At the end of this survey, GPs were offered to participate in a CATI in order to investigate selected topics in more detail. Only respondents interested provided contact details while the survey was anonymous.

### 2.2. Instruments

A web-based questionnaire was created based on a literature review about pandemic preparedness. A team of GPs, hospital physicians, nursing science and public-health researchers developed the questionnaire in an iterative process. The authors drafted the first version and refined it in multiple steps. The survey was piloted among 55 participants with the help of the platform unipark.com and finalised based on the preliminary results. Invitations were sent via e-mail. The survey was open from 17 March 2021 until 17 June 2021. One reminder was sent after four weeks. GPs without valid e-mail addresses, false e-mail addresses, and GPs in retirement were excluded from the population of GPs invited.

Items for the CATI were developed based on the free-text answers from the web-based survey. All GPs who had agreed to the telephone interviews were contacted by the Survey Center Bonn uzbonn—Society for empirical social research and evaluation in December 2021. The interviewers of uzbonn read the questionnaire to the participants to ensure the standardisation of the interview and recorded the results in a previously created input mask in accordance with the questionnaire. The duration of an interview ranged from 15–20 min. Appointments were rescheduled as necessary.

#### 2.2.1. Sociodemographic and Workplace Characteristics

**The web-based survey** collected sociodemographic characteristics about gender, years as a GP, and region of residence. Using the information of the participants’ German state, their location was summarized as one of the four regions ‘north’, ‘south’, ‘east’, and ‘west’. Lower Saxony, Schleswig Holstein, Bremen, and Hamburg were named as “North”. Bavaria and Baden-Württemberg were summed up as “South”. Mecklenburg Western Pomerania, Brandenburg, Berlin, Saxony-Anhalt, Saxony, and Thuringia were condensed to “East”. Hesse, Saarland, Rhineland Palatinate, and North-Rhine Westphalia were summarized as “West”. The survey asked further about workplace characteristics (number of personnel [self-employed GPs, employed GPs, GP trainees and practice assistants]; the number of practice personnel who tested positive for COVID-19; the number of patients seen per quarter; and usage of the German COVID tracing app [Corona-Warn-App]). 

**In the CATI**, respondents were asked about gender, years as a GP, and the number of practice personnel (self-employed GPs, employed GPs, GP trainees, and practice assistants). Regional clustering was performed in the same way as for the web-based survey.

#### 2.2.2. GPs’ Self-Perceived Leadership and Role in the Pandemic

**In the web-based survey**, the validated German translation [24] of the validated Crisis Leader Efficacy in Assessing and Deciding scale (C-LEAD) was used to assess self-perceived leadership [25]. It measures the efficacy of assessing information and decision-making in public health and safety crises [25]. The scale consists of nine items around information assessment and decision-making, which are measured on a 7-point rating scale from 1 (‘strongly disagree’) to 7 (‘strongly agree’). The average sum of all items represents the final score ranging from 7 to 63, with higher numbers representing higher self-perceived leadership [25]. Occupational workload due to the pandemic was measured based on a 5-point rating scale incorporating ‘very low’ to ‘very heavy’.

**In the CATI**, GPs’ experiences with their role in the pandemic, as well as their cooperation and communication with local health departments (LHD), were elicited based on ten statements and a 5-point rating scale (‘completely disagree’, ‘rather disagree’, ‘neutral’, ‘rather agree’, and ‘completely agree’). Similarly, the response options for some questions addressed factors influencing GPs’ satisfaction with working under pandemic circumstances (‘completely satisfied’, ‘rather satisfied’, ‘neutral’, ‘rather dissatisfied’, and ‘completely dissatisfied’). In order to obtain more detailed information, the interviewers asked open questions about additional factors driving satisfaction and dissatisfaction. GPs’ answers to the open questions were categorised by the interviewers based on predefined items. If the interviewers were not able to categorise an answer, it was recorded as free text. Items for satisfaction were ‘team work’, ‘support of my family’, ‘care of seriously ill patients’, ‘vaccinated many patients’, ‘high regard in the public’, ‘high appreciation from patients’, ‘doing something meaningful’, ‘contributing to patients’/public health’, and ‘staying healthy’. Items for dissatisfaction were ‘losing primary responsibility for my own patients’, ‘not knowing about regulations from public-health offices for my patients and their families’, ‘ambiguity about responsibilities for patients’, ‘GPs’ loss of power if local health departments take over’, ‘GPs can take more responsibilities and have the capacities to do so’, ‘increased costs due to hygienic measures’, ‘unexpected staff shortages’, ‘necessity of spatial patient separation’, ‘higher workload scheduling appointments’, ‘high workload in general’, and ‘death of own patients’. The C-LEAD scale was not used in the CATI.

#### 2.2.3. GPs’ Participation in New Services and Preferences for Future Pandemic Preparedness

**In the web-based survey**, GPs were asked if they had been involved in adapted and newly established regional services related to the pandemic, i.e., diagnostic centres, diagnostic teams, COVID-19 diagnostic practices, Corona-Taxis, COVID-19 outpatient clinics, COVID-19 focus practices, and vaccination centres/teams. Possible answer options were ‘I was involved’, ‘I am aware of the service but was not involved’, and ‘do not know’. The last two options were combined into ‘not involved’ for the purpose of this study. If GPs selected ‘involved’, they were asked to rate the service on a scale of 0–10 with 0 = very bad and 10 = very good. The question on participation in regional services was not linked to time periods, i.e., multiple participations could have happened sequentially or parallel, and they could have lasted from weeks to months. 

**In the CATI**, a binary question (yes/no) asked GPs whether they had been involved in adapted or newly established regional pandemic services, with the option of free-text answers for services not listed.

### 2.3. Statistics

The statistical significance was set at *p* < 0.05. Percentages and mean values were calculated for valid cases. Relative frequencies, percentages and standard deviations are calculated with respect to their sample sizes. The Net Promotor Score (NPS) [26] is based on categorising answers to the rating of services from 0 to 10. ‘Promoters’ are defined by a score of 9 or 10, ‘passives’ by 7 or 8, and ‘detractors’ range between 0 and 6. The NPS is calculated by the percentage of promoters minus the percentage of detractors, such that the NPS value ranges between minus 100 (not at all recommended) to plus 100 (strongly recommended).

Associations between GPs’ role satisfaction and GP characteristics (communication, cooperation and support with colleagues, politics, and especially LHDs) were analysed with Spearman’s correlation. An asymptotic Kruskal–Wallis test was conducted to assess associations between sociodemographic and workplace characteristics (independent variables) on leadership as measured by the C-LEAD score (dependent variable). Another asymptotic Kruskal–Wallis test analysed the sociodemographic and workplace characteristics as well as the C-LEAD score of GPs (independent variables) for participation to varying degrees in new services besides their practice (dependent variable). The Kruskal–Wallis tests were chosen instead of an ANOVA due to a missing Gaussian distribution of data. In order to perform the Kruskal–Wallis tests with different subgroups the following adjustments were made: years as a GP was divided into the subgroups ‘0–10’, ‘11–20’, ‘21–30’, and ‘>30’. The number of practice personnel was stratified into ‘0–3’, ‘4–6’, ‘7–9’, and ‘>10’. The number of practice personnel who tested positive for COVID was divided into five groups: ‘0%’, ‘>0–11.11%’, ‘>11.11–20%’, ‘>20–33.33%’, and ‘>33.33%’. The number of treated patients per quarter was subdivided into ‘up to 1000’, ‘1001–1500’, ‘1501–2000’, and ‘>2000’. Participation in new services was summarised to a total score per GP including ‘0’, ‘1–2’, and ‘2–7’.

Furthermore, the ten statements about GPs’ experiences with their role in the pandemic, as well as their cooperation and communication with LHDs, were tested for correlations with GPs’ role satisfaction. The scale levels of all items were ordinal, such that Spearman’s rank correlation was employed.

Participants were compared to nonparticipants by regional location and gender using a Chi-squared test (gender) and an ANOVA (region).

The statistical analysis was carried out using IBM^®^ SPSS^®^ Statistics for Windows version 26.0 (IBM Corp., Armonk, NY, USA). Free-text answers from the web-based survey were coded with MAXQDA 2021 (VERBI Software, Berlin, Germany: VERBI).

## 3. Results

### 3.1. Sociodemographic and Workplace Characteristics

Of the 10,600 GPs initially drawn in the sample, 9287 had a valid e-mail address and were currently working as GPs; 630 GPs completed the web-based survey (response rate: 6.8%). Respondents were mostly experienced (years as a GP: 18.8 ± 9.6). More than half of the participants were male (57.8%) and came from at least medium-sized practices (mean number of practice personnel: 8.0 ± 8.8; 50.4% treated at least 1500 patients/quarter). All regions were well represented. In the sample, 401 GPs (63.7%) stated that no COVID-19 cases occurred among their personnel. GPs reported a high workload (mean value 4.1 ± 0.8 from a maximum of five). Nearly 60% of GPs were using the German Corona-Warn-App for public contact tracing during the survey period. Table 1 shows the results for sociodemographic and workplace characteristics.

Overall, 37.9% of the GPs surveyed (*n* = 239 of 630) had volunteered for the CATI and were contacted. A total of 102 GPs were interviewed. The characteristics of the CATI respondents were similar to the overall sample. Their work experience was slightly higher (years as a GP: 20.1 ± 9.8 vs. 18.8 ± 9.6), while the percentage of male participants was slightly lower (55.8% vs. 57.8%), as was the average number of practice personnel (7.3 ± 8.5 vs. 8.0 ± 8.8).

The comparison between participants and nonparticipants revealed no group differences for the regional location (ANOVA, *p* = 0.126) but for gender (Chi-squared test, *p* < 0.001). Among the nonresponders, 18.1% were located in the North, 30.6% in the South, 21.6% in the East, and 29.7% in the West. The group of nonresponders had 4% more male participants.

### 3.2. GPs’ Self-Perceived Role and Leadership in the Pandemic

In the CATI, only 41.1% of the GPs were satisfied with their role in the pandemic. The majority agreed that their role had undergone changes in the pandemic (75.4%) and that politics underestimated the potential of GPs (77.5%). Regional networks were helpful for more than 60%. Nearly half the GPs (45.1%) reported feeling left alone. Although around 75% participated in at least one new service, only 25% indicated that their participation in new services was explicitly solicited. The experiences with LHDs were mixed: More than two-thirds of GPs described problems with communication and responsibility issues regarding patients and regulations. However, around 40% of GPs felt that LHDs played a vital role in disburdening them by contacting COVID-19-positive patients. Table 2 shows factors influencing GPs’ role with a focus on experiences with LHDs.

GPs’ role satisfaction with their role correlated positively with the use of their own regional networks (r = 0.239/*p* = 0.02), while it was negatively associated with the feeling of being left alone (r = 0.349/*p* < 0.001) and ambiguity regarding patient responsibilities between GPs and LHDs (r = 0.197). GPs’ belief that their role changed in the pandemic showed a tendency to correlate with their role satisfaction. There was no correlation between the number of new services in which GPs participated and GPs’ satisfaction (see Table 3).

Table 4 shows additional factors driving GPs’ satisfaction and dissatisfaction with the conditions of working under pandemic circumstances. The most frequently mentioned factors for satisfaction were a feeling of maintaining patients’ health (51.0%), doing something meaningful (44.1%), and feeling appreciated by patients (37.3%). Drivers for dissatisfaction included a higher workload in general (39.2%), a higher workload scheduling appointments (18.6%), and ambiguity about responsibilities for their patients (17.6%).

GPs’ leadership measured by the C-LEAD score showed high values (47.4 ± 8.5 from a maximum of 63) and correlated positively with patients seen per quarter (Kruskal–Wallis-H: 15.364, *p* = 0.002). GPs’ C-LEAD scores did not correlate with gender, professional experience, number of practice personnel, number of positive COVID-19 cases among the staff, region, or workload (see Table 5).

### 3.3. GPs’ Participation in New Services and Preferences for Future Pandemic Preparedness

The web-based survey showed GPs´ broad participation in newly established health services which was in addition to their practice duties: 299 GPs (47.5%) participated in one or two, while 158 GPs (25.0%) even in 3 to 7 new services. Similar results were obtained in the CATI. For details see Figure 1.

Most respondents in both groups participated in vaccination centres (web-based survey: 52.7%, CATI: 46.1%), followed by COVID-19 diagnostic practices (web-based survey: 37.8%, CATI: 42.0%). When asked for their recommendations for future pandemic services, respondents of the web-based survey rated COVID-19 focus practices and the Corona-Taxi best (NPS: 43.7 and 17.6, respectively), while diagnostic centres and diagnostic teams were rated worst (NPS: −31.0 and −22.8, respectively). Vaccination centres were among the three lowest-rated services. Participation rates in newly established health services were similar between the survey and CATI except for diagnostic centres (−14.2%) and COVID-19 focus practices (+20.0%) which reflects developments to different pandemic phases (survey: spring 2021; CATI: winter 2021). For details see Table 6.

A higher participation in newly established pandemic services was positively associated with the following sociodemographic and workplace characteristics: a higher number of practice personnel (H = 17.041/*p* = 0.001), a higher number of patients (H = 46.979/*p* < 0.001), a higher workload H = 12.136/*p* = 0.016), and a higher C-LEAD score (H = 23.031/*p* < 0.001). Practices with no and those with more than 33.3% COVID-19 cases among their personnel were less likely to participate in newly established services. Furthermore, GPs from the South and West of Germany were significantly more likely to participate in new pandemic services compared to those from the East or North. For details see Table 7.

## 4. Discussion

This representative mixed-methods study provides a snapshot of GPs’ self-perceived role, leadership, participation in newly established pandemic health services, and their preferences for future regional pandemic management at around months 15 and 22 of the pandemic (shortly after nationwide vaccination campaigns started). Although 72.5% of GPs were engaged not only as leaders in their practice but also in at least one newly established health service, only 41% were satisfied with their role in the pandemic and 77.5% believed that politics underestimated GPs’ potential. Germany’s current pandemic plan and its supplements, like that of many other countries, do not detail GPs’ role in the pandemic [27]. However, taking GPs’ attitudes and preferences into account in regional pandemic planning is crucial to maintaining and improving GPs’ cooperation in future pandemics [13].

Our respondents indicated low satisfaction with their own role in regional pandemic management, driven by the feeling of being left alone, feeling underappreciated, and role ambiguity. This is in line with results from the literature, where GPs report feeling ‘abandoned’ due to a strong focus on hospitals [9], being underappreciated [28,29,30] and having to fulfil unrealistic political promises [7]. The feeling of being underestimated and insufficiently involved seems to be well-founded since 84% of university hospitals in Germany led a structured regional cooperation, but only 20% built a structured cooperation with GPs [11]. Recent studies among Italian and German GPs showed that GPs felt insufficiently prepared to give patients appropriate clinical and organisational information about COVID-19 due to poor communication with local health authorities, a lack of clear and updated information, and a lack of time for self-education [8,31]. These feelings correspond to the low self-perceived pandemic preparedness among GPs in Germany [7,10,22] and other European countries [22,32]. In contrast, GPs in regions with prior infectious disease public health crises such as SARS and H1N1, e.g., Singapore, felt better prepared due to strengthened pandemic preparedness within the healthcare system [5]. In contrast to external factors driving dissatisfaction with working under pandemic conditions, GPs listed intrinsically motivating factors such as caring for their patients, contributing to society, and taking a leadership role as drivers for their satisfaction. This confirms findings from a previous study from Australia [33], Canada [34], and Europe [9,21]. A Europe-wide study identified governmental support, collaborations, and appreciation for their work as crucial for the wellbeing of GPs [21]. We did not find an association between role dissatisfaction and practice size or years of experience as a GP, which were reported as factors promoting distress in a study of 33 European countries [21].

Factors of satisfaction and dissatisfaction with working under pandemic circumstances are also in line with findings from prepandemic literature, where a high workload and missing appreciation of work are identified as dissatisfying [35]. Positive emotions, a sense of professional wellbeing, and the nature of the job were identified as maintaining GPs’ commitment [36], which is especially important during a public health crisis. The importance of satisfying factors, especially intrinsic motivation, is further enhanced by findings from a study of health personnel in Lithuania, where work meaningfulness shows a moderating effect between professional respect and performance outcomes, such that influencing work meaningfulness leads to increased performance [37]. These findings from the literature and our findings align with the job characteristics model of work motivation describing the relation between job characteristics and individual responses to work [38]. In the model, three psychological states are defined as experienced meaningfulness, responsibility for outcomes of work and knowledge of the actual work outcomes. These yield high internal work motivation, high-quality work performance, high satisfaction with the work, and low absenteeism and turnover as personal and work outcomes. Our most- indicated satisfying factors relate especially to the first two mentioned psychological states of the job characteristics model of work motivation [38] showing the high personal and work outcomes of GPs. However, the determination of causal mechanisms for satisfying factors remains a challenge [35].

Many GPs perceived themselves as leaders in their practices and embraced newly established or adapted services. Studies prior to the COVID-19 pandemic found that leaders of primary care clinics were likely to underestimate their role and potential impact [39]. Our finding of a C-LEAD score of 47.39 (SD: 8.51) for GPs is similar to published scores among crisis responders at federal agencies in the United States with 48.24 (SD: 6.57) [25], nurse practitioners with a Master’s degree with 48.26 (SD: 9.97) [40], and collegiate aviation crisis leaders with 47.65 (SD: 7.10) [41]. The achieved score indicates high resilience, motivation to lead in a crisis, and leader role-taking [25]. It shows that GPs were aware of their leading role and motivated to fulfil it, which should put them in a position to take on leadership roles in regional pandemic management as well. However, comparison of C-LEAD scores is challenging due to the lack of a cut-off value representing high leadership, in spite of extensive pretesting to ensure strong face validity and internal reliability [25]. The association between a high number of patients per quarter and a higher self-assessed leadership score is also interesting. The association does not seem to relate to the professional experience, since that characteristic exhibits no significant influence. Regarding the domains of the C-LEAD score, maintaining a high number of patients during the pandemic may indicate outstanding organizational and treatment skills, which may correlate with self-perceived leadership. Additionally, maintaining the treatment of many patients, while external conditions and workflows change, likely represents high resilience. Interestingly, gender, workload, practice size and region were not significant influences on GPs’ leadership.

The majority of our respondents participated in at least one service in addition to their daily practice. Factors associated with participating in more services included treating more patients, a higher workload, working in larger practices, and a higher self-perceived leadership. Our findings add to those of a recent survey among German GPs regarding Corona contact points (COVID-19-specialized primary care practices, outpatient infection centres and testing points), which strongly correlated with high intrinsic motivation and taking the initiative for opening a Corona contact point [42]. However, to our knowledge, the association with participation in multiple services has not been reported before.

GPs seemed more likely to prefer newly established services that were similar to existing services, such as the Corona-Taxi (similar to a home visit) or the COVID-19 focus practice (similar to a physician´s office). Such a mechanism of approval can be conceptualised as the ‘compatibility’ of the intervention with the broader context of work processes. In implementation science, this concept has previously been described as crucial for the adoption of innovations in primary care [43] and beyond [44]. Implementation science has also long posited that careful attention to the context in which innovations are implemented as well as clinician attitudes are crucial in the long-term adoption of new services [45,46]. This compatibility might be of particular importance in the high-anxiety environment of a beginning epidemic where many GPs felt ill-prepared [7,10,22,32,47]. Respondents were particularly dissatisfied with vaccination centres, even though almost half of them participated in them. Likely, they were also vaccinating in their own practices at this time [48]. Vaccination centres were meant to increase accessibility for all citizens [49], be time-efficient [50], and allow for a high throughput of inoculations [51]. Prior studies have reported high patient satisfaction with vaccination centres [52]. In contrast to our results, a French study of vaccination centres among patients and healthcare workers found that healthcare workers were very satisfied with their accessibility, hygiene, and confidentiality [53]. Our respondents might have been less satisfied with vaccination centres due to a double burden (the vaccination campaign started in practices during the survey period, while vaccination centres had already been operating for about four months in 2021). Alternatively, the negative perception might also reflect a perceived inefficient and costly setup since vaccination centres received 10× higher remuneration per vaccination than practices [54].

Furthermore, GPs’ experiences and satisfaction with respect to LHDs in the CATI indicated a challenging cooperation between GPs and LHDs. Information politics and communication of LHDs were broadly criticized by GPs. Both topics concerned not only GPs but patients as well since responsibilities were not clear to them. This ambiguity correlated negatively with GPs’ role satisfaction. GPs missed information about regulations and decisions, especially for their patients. Lacking cooperation with German LHDs and a lack of accessibility were also described by the recent literature [8]. However, in our study, this finding was not universal since half of the GPs felt disburdened by LHDs and the majority did not worry about losing competences to LHDs.

### Strengths and Limitations

This mixed-method study with quantitative and qualitative data provides a nuanced view of respondents’ perceptions. For the web-based survey, participants were selected in a multilevel clustered sampling to assure representativeness. However, our response rate was low at 6.8% but in line with other pandemic studies among German GPs [2,10]. The regional location between participants and nonparticipants revealed no selection bias. Differences in gender distribution were significant, such that we cannot rule out the possibility of an overrepresentation of male GPs. Furthermore, 37.9% of the survey participants volunteered for the CATI, which is a remarkably high rate. The results of this study need to be interpreted as a snapshot of months 15 and 22 of the pandemic. Additionally, the factors influencing satisfaction and dissatisfaction should be interpreted against the background of conditions of working under pandemic circumstances and do not represent the participants’ general job satisfaction.

## 5. Conclusions

GPs were less than satisfied with their role in the pandemic, despite seeing themselves as leaders. They participated extensively in newly established and adapted services, but perceived services unrelated to their practices as suboptimal. Their high intrinsic motivation originating in the nature of their work and appreciation by their patients was important for them working under pandemic circumstances, but they felt mostly left out of planning and management. Hence, politics and public institutions need to pay attention to the situation of GPs and include them in the organization of ambulatory healthcare pandemic preparedness. This implies also that the cooperation and communication between GPs and LHDs require a clear and stable concept. Future research should focus on ways to integrate GPs’ perspective into pandemic preparedness planning and on how best to ensure that new or adapted services are compatible with existing services. Hopefully, the findings from this study will be useful in these efforts.

## Figures and Tables

**Figure 1 ijerph-20-06088-f001:**
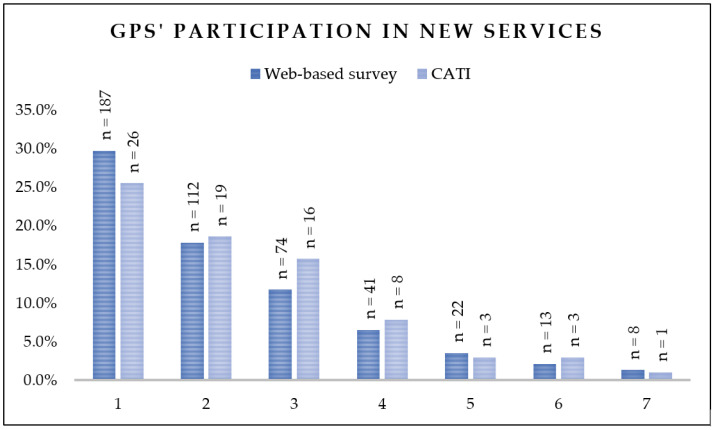
Number of newly established health services in which GPs’ participated (survey: *n* = 630, CATI *n* = 102).

**Table 1 ijerph-20-06088-t001:** Personal and workplace characteristics of GPs (survey: *n* = 630, CATI: *n* = 102).

	Survey (*n* = 630)			CATI (*n* = 102)		
	N	%	Missing Values (%)	N	%	Missing Values (%)
Gender	630	100	0 (0)	102	100	0 (0)
Male	364	57.8		57	55.8	
Female	263	41.7		45	44.2	
Neutral	3	0.5		0	0	
Years as a GP	600	95.2	30 (4.8)	102	100	0 (0)
0–10	138	21.9		20	19.6	
11–20	205	32.5		36	35.3	
21–30	177	28.1		32	31.4	
>30	80	12.7		14	13.7	
Number of practice personnel	622	98.7	8 (1.3)	102	100	0 (0)
0–3	144	22.9		32	31.4	
4–6	204	32.8		31	30.4	
7–9	115	18.3		17	16.7	
>9	159	25.2		22	21.6	
Region	630	100	0 (0)	90	88.2	12 (11.8)
North	104	16.5		20	21.7	
South	173	27.5		17	18.5	
East	127	20.2		25	27.2	
West	226	35.9		30	32.6	

**Table 2 ijerph-20-06088-t002:** CATI: GPs’ role in the pandemic and experiences with local health departments (*n* = 102).

Item	Satisfied (%)	Neutral (%)	Dissatisfied (%)
GPs’ satisfaction with their role in the pandemic	42 (41.1)	36 (35.3)	24 (23.5)
	**Agree (%)**	**Neutral (%)**	**Disagree (%)**
The role of GPs changed during the pandemic	77 (75.4)	12 (11.8)	13 (12.7)
Feeling of being left alone in the pandemic	46 (45.1)	28 (27.5)	28 (27.5)
Politics underestimated the potential of GPs	79 (77.5)	17 (16.7)	6 (5.9)
The use of own regional networks was helpful	65 (63.8)	17 (16.7)	12 (11.7)
GPs were requested for many new services	26 (25.5)	30 (29.4)	46 (45.1)
GPs were not informed by local health departments about regulations for their patients	80 (78.4)	12 (11.8)	9 (8.8)
It was not evident for patients whether GPs or local health departments were in charge	71 (69.6)	16 (15.7)	13 (12.7)
Local health departments made decisions without informing GPs	72 (70.6)	11 (10.8)	18 (17.6)
GPs were disburdened by local health departments contacting COVID-19-positive patients	42 (41.2)	17 (16.7)	41 (40.2)
Poor information policy of local health departments	66 (64.7)	21 (20.6)	14 (13.8)

**Table 3 ijerph-20-06088-t003:** CATI: Factors influencing GPs’ satisfaction with working under pandemic circumstances (*n* = 102).

Items	Correlation Coefficient r	Number of GPs
The role of GPs changed during the pandemic	−0.181	102
Feeling of being left alone in the pandemic	−0.349	102
Politics underestimated the potential of GPs	0.062	102
The use of own regional networks was helpful	0.239	94
GPs were requested for many new services	0.092	102
GPs were not informed by local health departments about regulations for their patients	−0.121	101
It was not evident for patients whether GPs or local health departments were in charge	−0.197	100
Local health departments made decisions without informing GPs	−0.058	101
GPs were disburdened by local health departments contacting COVID-19-positive patients	0.092	100
Poor information policy of local health departments	−0.141	101
Number of participations in new services	−0.04	102

**Table 4 ijerph-20-06088-t004:** CATI: Additional factors driving GPs’ satisfaction and dissatisfaction with working under pandemic circumstances (*n* = 102).

Satisfaction	*n*	%
Contributing to patients’/public health	52	51.0
Doing something meaningful	45	44.1
High appreciation from patients	38	37.3
Administered many vaccinations	36	35.3
High regard in the public	25	24.5
Teamwork	21	20.6
I have not been ill	16	15.7
Care of seriously ill patients	13	12.7
Support of my family	11	10.8
**Dissatisfaction**	** *n* **	**%**
Higher workload in general	40	39.2
Higher workload scheduling appointments	19	18.6
Ambiguity about responsibilities for patients	18	17.6
Not knowing about regulations from local health departments regarding my patients and their families	13	12.7
Increased costs due to hygienic measures	13	12.7
Necessity of spatial patient separation	11	10.8
GPs can take more responsibilities and have the capacities to do so	9	8.8
Unexpected staff shortages	8	7.8
Death of own patients	6	5.9
Losing primary responsibility for own patients	5	4.9
GPs’ loss of power if local health departments take over	4	3.9

**Table 5 ijerph-20-06088-t005:** Web-based survey: Associations between workplace characteristics and GPs’ leadership as measured by C-LEAD (Kruskal–Wallis test) (*n* = 630).

Characteristic	Kruskal–Wallis-H	Degrees of Freedom	Significance (*p* < 0.05)
Gender	2.177	2	0.337
Years as a GP	1.423	3	0.700
Number of practice personnel	4.688	3	0.196
Number of practice personnel tested positive for COVID-19	4.411	4	0.353
Region	0.964	3	0.810
Patients per quarter	15.364	3	0.002
Workload	6.756	4	0.149

**Table 6 ijerph-20-06088-t006:** GPs’ participation in newly established health services and their recommendations for future pandemic management (survey: *n* = 630, CATI *n* = 102).

	Involved (Survey)		Perception (Survey)	Involved (CATI)	
Services	*n*	%	NPS	*n*	%
COVID-19 diagnostic practices	238	37.8	8.7	42	41.2
Diagnostic centres	145	23.0	−31.0	9	8.8
Diagnostic teams	91	14.4	−22.8	16	15.7
Corona-Taxi	34	5.4	17.6	13	12.8
COVID-19 focus practices	103	16.3	43.7	37	36.3
COVID-19 outpatient clinics	98	15.6	−8.2	20	19.6
Vaccination centres/teams	332	52.7	−13.3	47	46.1

**Table 7 ijerph-20-06088-t007:** Web-based survey: GP and practice characteristics associated with participation in more newly established services (*n* = 630).

Characteristic	Kruskal–Wallis-H	Degrees of Freedom	Significance (*p* < 0.05)
Patients per quarter	46.979	3	<0.001
C-LEAD score	23.031	3	<0.001
Number of practice personnel	17.041	3	0.001
Region	15.373	3	0.002
Number of practice personnel tested positive for COVID-19	12.275	4	0.015
Workload	12.136	4	0.016
Years as a GP	5.386	3	0.146
Gender	2.406	2	0.300

## Data Availability

The datasets used and analysed are available on reasonable request from the corresponding author.

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
