# Peer review of "German GPs’ Self-Perceived Role in the COVID-19 Pandemic: Leadership, Participation in Regional Services and Preferences for Future Pandemic Preparedness"

_ijerph, 2023, doi:10.3390/ijerph20126088_

Round 1

Reviewer 1 Report

No major criticism to be expressed

Minor criticism: details on how evaluation scales are defined and how questionnaires were developed and administered would have been appreciated, may be as supplemental material.

Reviewer 2 Report

The study has weaknesses and points that can be improved, on which I suggest to intervene to improve its quality.

First of all, the response rate is very low (102 interviews/630 questionnaires/9737 doctors). The hypotheses relating to the relationship between the reasons for participating in the study and the variables being studied should be investigated. For example, it is precisely the doctors who have participated the most in local structures could be the ones most available to respond to a survey. This risk should be discussed and if possible controlled.

Second, the study appears overly descriptive. Given the presence of different possible factors of satisfaction or dissatisfaction, we do not see an effort to organize these factors around different causal mechanisms. The authors could try to better study the structure of the data describing these factors by revealing clues about the type of underlying factors. For example, the organizational cultures of the different services and of the networks with which doctors collaborate, the type of patient orientation that characterizes them, the vision of the doctor's role, etc. A better systematization of the factors and the relationship between the items could better bring out different profiles of the medical role or different organizational cultures and service concepts present in the territorial networks. In the absence of this, there is a risk of giving an interpretation of satisfaction as deriving from personal factors and perceptions.

This systematization would have been facilitated if the proposed factors had been organized, on the basis of the literature, on motivation and job satisfaction factors, covering all possible aspects and defining them in a broad way, while 'family support', 'teamwork ', or the 'number of services attended', thus expressed, do not allow us to have clues about the causal mechanism.

Finally, it is not clear why the relationship between the C-LEAD index and the other variables has not been developed. This would be a strong point of the work. It would be interesting to verify the connection between the self-efficacy of leadership in the crisis, satisfaction and the families of factors that qualify the experience of participation and collaboration. For example, high leadership may generate dissatisfaction if services are uncooperative, while poor leadership may not generate dissatisfaction in the same condition. I suggest exploring as much as possible this relationship between the level of self-efficacy of the leadership in the crisis and the type of organizational culture (more or less collaborative) of the network, looking for possible combinations. Strong correlations may also emerge that are not visible without control variables.

Reviewer 3 Report

The study describes German GPs' self-perceived role in the COVID-19 pandemic, including leadership, participation in regional services and preferences for future pandemic preparedness.

The study is useful for future pandemic response.

The paper would be even better if the following six points were improved.

1. Satisfaction seems to be one of the key words in the paper as a whole. Please consider if you can include satisfaction in the title. If the author, on second thought, does not wish to do so, the author need not necessarily do so.

2. In the introduction section, please explain why you focused on 1) leadership, 2) participation in regional services, and 3) preferences for future pandemic preparedness. This would improve the paper.

3.In the Methods section, please clarify whether the C-LEAD scale is also reliable and validated in German.

4. In the discussion, please explain in more detail why a C-LEAD score of 47.4 is considered high.

5. In any tables, please spell out the abbreviations so that the reader can understand the abbreviations by looking at the table alone.

6. In the limitations of the study, it would be good to describe the possibility that the questions regarding satisfaction with the GP role do not necessarily represent satisfaction.

Reviewer 4 Report

Thank your for giving me the opportunity to review this paper. It is a quite short and compact paper that is easy to read.

I think that the paper should be developed with a clear aim. Without a clear aim it is difficult to present a clear and sufficient conclusion. Therefore, I would like the authors to also be more clear about the conclusions.

In the en of the first paragraph in the materials and methods section the authors write about a "majority of GPs"... and also "mainly GP owned ambulatory health care centres". I think this description should be clearer. Is the GP owned general practices mainly GP owned ambulatory health care centres?

The leadership seems to be included as an aspect. However, it is difficult to really understand what the authors really relate to the leadership aspect. May the leadership aspect could be removed.

Round 2

Reviewer 3 Report

I have seen the revised manuscript. The six points I indicated have been improved. I think it is a good manuscript.